# Beyond Differentiability: Neurosymbolic Learning with Black-Box Programs

## Abstract

Neurosymbolic learning has demonstrated promising potential as a paradigm to combine the worlds of classical algorithms and deep learning. However, existing general neurosymbolic frameworks require that programs be written in differentiable logic programming languages, restricting their applicability to a small fragment of algorithms. We introduce Infer-Sample-Estimate-Descend (ISED), a general algorithm for neurosymbolic learning with black-box programs. We evaluate ISED extensively on a set of 30 benchmark tasks that encompass rich data types and reasoning patterns. ISED achieves 30% higher accuracy than end-to-end neural baselines. Moreover, ISED's solutions often outperform those obtained using Scallop, a state-of-the-art neurosymbolic framework: the programs in 17 (61%) of the benchmarks cannot be specified using Scallop, and ISED on average achieves higher accuracy on those that can be specified using Scallop.

## 1 Introduction

Neurosymbolic learning (Chaudhuri et al., 2021) is an emerging paradigm that combines the otherwise complementary worlds of classical algorithms and deep learning. On the one hand, classical algorithms are suitable for exactly-defined tasks with structured data, such as sorting a list of numbers. On the other hand, deep learning is suitable for tasks involving unstructured data that cannot be solved by classical algorithms, such as image recognition. Neurosymbolic learning combines the strengths of these two approaches to achieve better performance, interpretability, and robustness.

State-of-the-art approaches to neurosymbolic learning have demonstrated high accuracy on tasks such as interpretable reinforcement learning Dutta et al. (2023), visual question answering (Huang et al., 2021), and Sudoku solving (Wang et al., 2019). However, these approaches require programs—the symbolic components of a neurosymbolic model—to be differentiable. As such, these programs must be specified in languages like Datalog (Li et al., 2023) or Prolog (Manhaeve et al., 2018), which are less expressive or accessible than general-purpose languages like Python.

In this paper, we propose a neurosymbolic learning approach based on forward evaluation, which is compatible with arbitrary black-box programs that can be written in general-purpose languages. Our approach, called ISED (Infer-Sample-Estimate-Descend), yields a framework that expands the applicability of neurosymbolic learning by using sampling instead of differentiable logics. ISED targets the setting of *algorithmic supervision* (Petersen et al., 2021b), wherein the black-box program $P$ is applied to the output of a neural model $M_\theta$, and the goal is to optimize the model parameter $\theta$ using only supervision on end-to-end labels $y = P(M_\theta(x))$. The main challenge in this setting concerns how to compute the loss across the black-box program.

ISED is used during training and is composed of four phases. In the *Infer* step, neural models predict probabilistic distributions for each input. In the *Sample* step, ISED samples inputs from each distribution according to a given sampling strategy, and executes the black-box program on the sampled inputs. In the *Estimate* step, ISED estimates probabilities and their gradients using the obtained set of input-output pairs. Lastly, in the *Descend* step, inputs that yield outputs that are equal (resp. dissimilar) to the ground truth are rewarded (resp. penalized) in the loss function.

We evaluate ISED on a set of 30 benchmark tasks wherein the black-box programs are written in Python. The tasks cover a wide range of data types—including both synthetic and real-world image data—and programs with rich reasoning patterns, including programs taken from the Leetcode

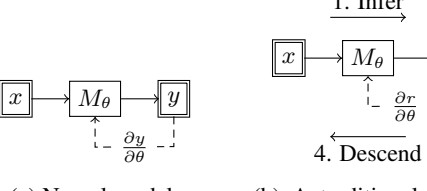 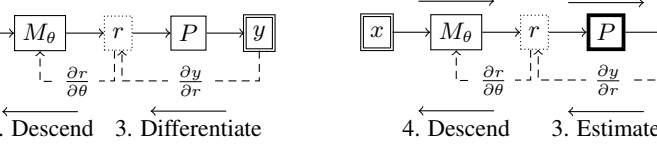

(a) Neural model.

(b) A traditional neurosymbolic program, where $\frac{\partial y}{\partial r}$ is computed by differentiating white-box $P$.

(c) A neurosymbolic program with black-box component, where $\frac{\partial y}{\partial r}$ is computed by sampling and evaluation.

Figure 1: Comparison of different paradigms. Program $P$ operates on structured input $r$ whereas neural model $M_\theta$ with parameter $\theta$ operates on unstructured input $x$. Under *algorithmic supervision*, neurosymbolic learning approaches must optimize $\theta$ without supervision on $r$.

programming platform. We compare the accuracy of ISED to end-to-end neural baselines and to Scallop (Li et al., 2023), a state-of-the-art neurosymbolic framework wherein programs are written in a restricted logic programming language, Datalog.

In summary, the main contributions of this paper are as follows: 1) we introduce a general algorithm for learning with black-box symbolic programs, 2) we implement a general learning pipeline that integrates well with different types of structured and unstructured data, 3) we propose a diverse set of benchmarks to evaluate our algorithm, and 4) we conduct a thorough evaluation using existing neurosymbolic techniques against these benchmarks.

## 2 OVERVIEW

### 2.1 PROBLEM STATEMENT

Neurosymbolic learning combines the paradigms of classical algorithms and deep learning. Classical algorithms, like logic programs, operate only on structured data. On the other hand, neural models operate on structured or unstructured data, and can be differentiated, as shown in Fig. 1a.

In the traditional neurosymbolic learning setting, we attempt to optimize model parameters $M_\theta$ which is being algorithmically supervised by a fixed program $P$. Typically, there exists some mechanism for automatic differentiation of programs, i.e., computing $\frac{\partial y}{\partial r}$ in Fig. 1b. For example, Scallop (Li et al., 2023) uses provenance semirings to approximate the probabilities of outputs $y$ given the predicted distributions for structured inputs $r$ computed by the neural models. Different semirings correspond to different heuristics for approximating the gradient $\frac{dy}{dr}$ of the output of a logic program. Calculating these gradients relies on two key assumptions: (1) programs are white boxes, and (2) programs are written in a differentiable manner, typically in a logic programming language.

We aim to find a general algorithm for computing $\frac{\partial y}{\partial r}$ without assumptions (1) and (2), as shown in Fig. 1c. The important distinction, compared to the traditional neurosymbolic learning setting, is that $P$ cannot be differentiated, meaning $\frac{\partial y}{\partial r}$ can only be approximated using forward execution. Our key insight is that no matter how complex a given program is, we can execute it many times on many combinations of structured inputs. We can then obtain outputs that, when compared to the ground truth, reveal information about which inputs were likely correct. We now introduce three motivating applications (Fig. 2) that can be framed in this setting, namely hand-written formula evaluation (HWF), sorting MNIST digits, and determining severity of disease in coffee leaves.

### 2.2 MOTIVATING EXAMPLES

**Hand-written Formula.** In this task from Li et al. (2020), a model is given a list of hand-written symbols containing digits (0-9) and arithmetic operators ($+$, $-$, $\times$, and $\div$). The dataset contains length 1-7 formulas free of syntax error or divide-by-zero errors. The model is trained with supervision on the evaluated result (floating-point number) without the intermediate label of each symbol. Since inputs are combinatorial and the results are rational numbers, end-to-end neural methods struggle to produce accurate results. Meanwhile, neurosymbolic methods for this task either use specialized algorithms Li et al. (2020) or handcraft a differentiable program Li et al. (2023).

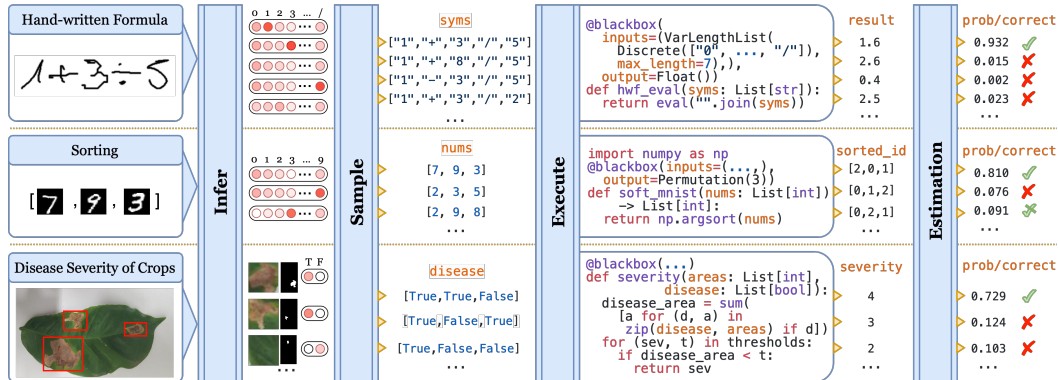

Figure 2: Illustration of our inference pipeline for three example applications.

With ISED, the program can be written in just a few lines of Python (Fig. 2). It takes in a list of strings representing symbols, and simply invokes the Python `eval` function on the joined expression string. Using the `blackbox` decorator provided by ISED, the program is configured to take in inputs of variable-length lists (with maximum length 7), with each element being a string representing a symbol. Under ISED, the decorated `hwf_eval` function can be used just like any other PyTorch (Paszke et al., 2019) module, and will internally perform sampling and probability estimation to facilitate neurosymbolic learning. Compared to Li et al. (2020) and Li et al. (2023), our solution achieves comparable performance while being more intuitive to machine learning programmers.

**Sorting MNIST Digits.** We next consider the heavily studied example of sorting. We are specifically interested in the task of computing the ordering of indices that would sort a list of three hand-written digits from the MNIST dataset (Lecun et al., 1998). One solution is specialized differentiable sorting networks (Petersen et al., 2021a) while other solutions, like DeepProbLog (Manhaeve et al., 2018), are general neurosymbolic learning frameworks that are applicable to sorting tasks.

ISED provides another solution to sorting, offering more generality than differentiable sorting networks, higher accuracy than DeepProbLog, and the unique ability to use programs written in any language such as Python. Specifically, our `sort_mnist` program is configured to take in a list of digits (0-9) and return a sorted indices array, expressed as `Permutation`. The function simply calls `argsort` provided by the Numpy library. Compared to a differentiable language like Deep-ProbLog, ISED provides a more idiomatic way of programming for machine learning programmers.

**Determining Disease Severity in Crops.** We finally consider a real-world example that deals with the problem of classifying disease severity in crops. In this task, the model must determine the severity (1-5, from low to high) of the disease (Fig. 2), indicated by the proportional area of *rust* on each leaf. Images of leaves are provided in a dataset by Brito Silva et al. (2020). Traditional neural methods take the whole image and predict the severity directly, without explicit notion of rust areas, resulting in solutions that are data-inefficient, inaccurate, and harder to understand and debug.

We can instead solve the problem using a neurosymbolic solution. Here, the neural model takes the whole image, segment it into patches, and classify each patch into rust or non-rust. The symbolic component then sums the area of the patches that are identified as rust, and determines the severity based on thresholds. With this method, we have a pipeline that explicitly learns about rust and determines the severity based on rust areas. Using the pre-trained Segment Anything Model (SAM) (Kirillov et al., 2023) and a randomly initialized rust classifier, our neurosymbolic solution outperforms the end-to-end neural approach by 16% while also providing better interpretability.

## 3 METHODOLOGY

### 3.1 PRELIMINARIES AND PROGRAMMING INTERFACE

ISED allows programmers to write black-box programs that operate on diverse structured inputs and outputs. In order for such programs to interact with neural networks, we define an interface named *structural mapping*. This interface serves to (1) define the data-types of black-box programs' input and output, (2) marshall and un-marshall data between neural networks and logical black-box

| Mapping ($\tau$) | Set Interpretation (SET($\tau$)) | Tensor Interpretation (DIST($\tau$)) |
|---|---|---|
| DISCRETE$_\Sigma$ | $\Sigma$ | $\{\vec{v} \mid \vec{v} \in \mathbb{R}^{|\Sigma|}, v_i \in [0,1], i \in 1 \dots |\Sigma|\}$ |
| FLOAT | $\mathbb{R}$ | n/a |
| PERMUTATION$_n$ | $\{\rho \mid \rho \text{ is a permutation of } [1, \dots, n]\}$ | $\{[\vec{v_1}, \dots, \vec{v_n}] \mid \vec{v_i} \in \mathbb{R}^n, v_{i,j} \in [0,1], i \in 1 \dots n\}$ |
| TUPLE($\tau_1, \dots, \tau_m$) | $\{(a_1, \dots, a_m) \mid a_i \in \text{SET}(\tau_i)\}$ | $\{(a_1, \dots, a_m) \mid a_i \in \text{DIST}(\tau_i)\}$ |
| LIST$_n(\tau')$ | $\{[a_1, \dots, a_j] \mid j \leq n, a_i \in \text{SET}(\tau)\}$ | $\{[a_1, \dots, a_j] \mid j \leq n, a_i \in \text{DIST}(\tau)\}$ |

Table 1: Set and tensor interpretations of different structural mappings.

| Mapping ($\tau$) | Vectorizer ($\delta_\tau(y, \hat{y})$) | Aggregator ($\sigma_\tau(\hat{r}, \hat{p})$) |
|---|---|---|
| DISCRETE$_n$ | $\boldsymbol{e}^{(y)}$ with dim $n$ | $\hat{p}[\hat{r}]$ |
| FLOAT | $[\mathbf{1}_{y=\hat{y}_i} \text{ for } i \in [1, \dots, \text{length}(\hat{y})]]$ | n/a |
| PERMUTATION$_n$ | $[\delta_{\text{DISCRETE}_n}(y[i]) \text{ for } i \in [1, \dots, n]]$ | $\otimes_{i=1}^n \sigma_{\text{DISCRETE}_n}(\hat{r}[i], \hat{p}[i])$ |
| TUPLE($\tau_1, \dots, \tau_m$) | $[\delta_{\tau_i}(y[i]) \text{ for } i \in [1, \dots, m]]$ | $\otimes_{i=1}^m \sigma_{\tau_i}(\hat{r}[i], \hat{p}[i])$ |
| LIST$_n(\tau')$ | $[\delta_{\tau'}(a_i) \text{ for } a_i \in y]$ | $\otimes_{i=1}^n \sigma_{\tau'}(\hat{r}[i], \hat{p}[i])$ |

Table 2: Vectorize and aggregate functions of different structural mappings.

functions, and (3) define the loss. We define a *structural mapping* as either a discrete mapping (with $\Sigma$ being the set of all possible elements), a floating point, a permutation mapping with $n$ possible elements, a tuple of mappings, or a list of upto $n$ elements. We define $\tau$ inductively as follows:

$$\tau ::= \text{DISCRETE}(\Sigma) \mid \text{FLOAT} \mid \text{PERMUTATION}_n \mid \text{TUPLE}(\tau_1, \dots, \tau_m) \mid \text{LIST}_n(\tau)$$

With this, we further define $\text{INTEGER}_j^k = \text{DISCRETE}(\{j, \dots, k\})$, $\text{DIGIT} = \text{INTEGER}_0^9$, $\text{ALPHA} = \{0, \dots, 9, A, \dots, Z, a, \dots, z\}$, and $\text{BOOL} = \text{DISCRETE}(\{\text{true}, \text{false}\})$. We also define a *black-box program* $P$ as a function $(\tau_1, \dots, \tau_m) \to \tau_o$, where $\tau_1, \dots, \tau_m$ are the input types and $\tau_o$ is the output type. For example, the structured input mapping for the hand-written formula task is $\text{LIST}_7(\text{DISCRETE}(\{0, \dots, 9, +, -, \times, \div\}))$, and the structured output mapping is FLOAT. The mapping suggests that the program takes a list of length 7 as input, where each element is a digit or arithmetic operator, and returns a floating point number.

There are two interpretations of a structured mapping: the set interpretation SET($\tau$) and the tensor interpretation DIST($\tau$). The set interpretation represents a mapping with defined values, e.g., a digit with value 8. The tensor interpretation represents a mapping where each value is associated with a probability distribution, e.g., a digit that is 1 with probability 0.6 and 7 with probability 0.4. These two interpretations are defined for the different structural mappings in Table 1.

In order to represent the ground truth output as a distribution mapping to be used in the loss function, there needs to be a mechanism for transforming SET($\tau$) output mappings into DIST($\tau$) output mappings. For this purpose, we define a *vectorize* function $\delta_\tau : (\text{SET}(\tau), 2^\tau) \to \text{DIST}(\tau)$ for the different output mappings $\tau$ in Table 2. When considering a datapoint $(x, y)$ during training, ISED samples many inputs and obtains a list of outputs $\hat{y}$. The vectorizer then takes the ground truth $y$ and the outputs $\hat{y}$ as input and returns the equivalent distribution-interpretation mapping of $y$. While $\hat{y}$ is not used by $\delta_\tau$ in most cases, we include it as an argument so that FLOAT output mappings can be discretized, which is necessary for vectorization. For example, if the inputs to the vectorizer for the hand-written formula task are $y = 2.0$ and $\hat{y} = [1.0, 3.5, 2.0, 8.0]$, then it would return $[0, 0, 1, 0]$.

We also require a mechanism for aggregating the probabilities that resulted in a particular output. With this aim, we define an *aggregate* function $\sigma_\tau : (\text{SET}(\tau), \text{DIST}(\tau)) \to \mathbb{R}$ for different input mappings $\tau$ in Table 2. ISED aggregates probabilities either by taking their minimum or their product, and we denote both operations by $\otimes$. The aggregator takes as input sampled inputs $\hat{r}$ and neural predictions $\hat{p}$ from which $\hat{r}$ was sampled. It gathers values in $\hat{p}$ at the indices in $\hat{r}$ and returns the result of $\otimes$ applied to these values. For example, if the aggregator for the hand-written formula task has $\otimes$ specified as min and takes $\hat{r} = [1, +, 1]$ and $\hat{p}$ as input where $\hat{p}[0][1] = 0.1$, $\hat{p}[1][+] = 0.05$, and $\hat{p}[2][1] = 0.1$ then it would return 0.05.

## 3.2 ALGORITHM

We now formally state the ISED algorithm. For a given task, there is a black-box program $P$, taking $m$ inputs, that operates on structured data. Let $\tau_1, \dots, \tau_m$ be the mappings for these inputs and $\tau_o$ the mapping for the program's output. We write $P$ as a function from its input mappings to its output

mapping: $P : (\tau_1, ..., \tau_m) \to \tau_o$. For each unstructured input $i$ to the program, there is a neural model $M_{\theta_i}^{(i)} : x_i \to \tau_i^{\text{SET}}$. $S$ is a sampling strategy (e.g., categorical sampling), and $k$ is the number of samples to take for each training example. There is also a loss function $\mathcal{L}$ whose first and second arguments are the prediction and target values respectively. We state the algorithm in pseudocode in Algorithm 1 and also present its steps with the hand-written formula task:

---

**Algorithm 1** ISED training pipeline

---

    $P$ is black-box program $(\tau_1, \ldots, \tau_m) \to \tau_o$, $M_{\theta_i}^{(i)}$ a neural model $x_i \to \tau_i^{\text{SET}}$ for each $\tau_i$, $S$ is sampling strategy, $k$ is sample count for each training example, $\mathcal{L}$ is loss function, $\mathcal{D}$ the dataset
1: **procedure** TRAIN
2:    **for** $((x_1, \ldots x_m), y) \in \mathcal{D}$ **do**
3:        **for** $i \in 1 \ldots m$ **do**
4:            $\hat{p}[i] \leftarrow M_{\theta_i}^{(i)}(x_i)$                                        ▷ **Infer**
5:        **end for**
6:        **for** $j \in 1 \ldots k$ **do**
7:            **for** $i \in 1 \ldots m$ **do**                                     ▷ **Sample**
8:                Sample $\hat{r}_j[i]$ from $\hat{p}[i]$ using $S$         ▷ Use $S$ to sample from the distributions
9:            **end for**
10:       $\hat{y}_j \leftarrow P(\hat{r}_j)$                         ▷ Compute output of $P$ with inputs $\hat{r}_j$
11:       **end for**
12:       $\hat{w} \leftarrow$ normalize$([\omega(y_i, \hat{y}, \hat{r}, \hat{p})$ for $y_i \in \tau_o$ (or $y_i \in \hat{y})])$         ▷ **Estimate**
13:       $l \leftarrow \mathcal{L}(\hat{w}, w)$
14:       Compute $\frac{\partial l}{\partial \theta}$ by performing back-propagation on $l$
15:       Optimize $\theta$ based on $\frac{\partial l}{\partial \theta}$ using an optimizer (e.g. Adam Optimizer)      ▷ **Descend**
16:    **end for**
17: **end procedure**

---

**Infer.** The training pipeline starts with an example from the dataset, $(x, y) = ([\wedge, +, 2], 3.0)$, and uses a CNN to predict these images, as shown on lines 3-4. ISED initializes $\hat{p} = M_\theta(x)$.

**Sample.** ISED samples $\hat{r}$ from $\hat{p}$ for $k$ iterations using strategy $S$. For each sample $j$, the algorithm initializes $\hat{r}_j$ to be the structured data sampled from $\hat{p}$, as shown on lines 6-8. To continue our example, suppose ISED initializes $\hat{r}_j = [7, +, 2]$ for sample $j$. The next step is to execute the program on $\hat{r}_j$, as shown on line 10, which in this example means setting $\hat{y}_j = P(\hat{r}_j) = 9.0$.

**Estimate.** In order to compute the prediction value to use in the loss function, ISED must consider each output $y_i$ in the output mapping and accumulate the aggregated probabilities for all sampled inputs that resulted in output $y_i$. We specify that $\otimes$ is the `min` function in this example, and we also specify $\otimes$ as the `max` function. Note that ISED requires that $\otimes$ and $\oplus$ represent either `min` and `max` respectively or multiplication and addition respectively. We define an *accumulate* function that takes as input an element of the output mapping $y_i$, sampled outputs $\hat{y}$, sampled inputs $\hat{r}$, and predicted input distributions $\hat{p}$. The accumulator performs the $\oplus$ operation on aggregated probabilities for elements of $\hat{y}$ that are equal to $y_i$ and is defined as follows:

$$\omega(y_i, \hat{y}, \hat{r}, \hat{p}) = \oplus_{j=1}^{k} \mathbf{1}_{\hat{y}_j = y_i} \sigma_{\tau_o}(\hat{r}_j, \hat{p}_j)$$

ISED then sets $\tilde{w} = [\omega(y_i, \hat{y}, \hat{r}, \hat{p})$ for $y_i \in \tau_o]$ in the case where $\tau_o$ is not FLOAT. When $\tau_o$ is FLOAT, which is the case for hand-written formula, it only considers $y_i \in \hat{y}$. Next, ISED normalizes $\tilde{w}$ by performing $L_2$ normalization over each element in $\tilde{w}$ and sets $\hat{w}$ to this result. ISED then initializes $l = \mathcal{L}(\hat{w}, w)$ and computes $\frac{\partial l}{\partial \theta_i}$ for each input $i$. These steps are shown in lines 12-14.

**Descend.** The last step is shown on line 15, where the algorithm optimizes $\theta_i$ for each input $i$ based on $\frac{\partial l}{\partial \theta_i}$ using an optimizer. This completes the training pipeline for one example, and the algorithm returns all final $\theta_i$ after iterating through the entire dataset.

## 4   BENCHMARK SUITE

In this work, we present a new benchmark suite to evaluate neurosymbolic frameworks. It is composed of 30 tasks, including 1) synthetic tasks curated using MNIST-R benchmarks and Leetcode

| Suite | Task | Input $(\tau_1, \ldots, \tau_m)$ | Output $(\tau_o)$ | Perceptual Dataset |
|---|---|---|---|---|
| MNIST-R / SVHN-R | $\text{sum}_n$, where $n \in \{2, 3, 4\}$ | $\text{DIGIT}^n$ | $\text{INTEGER}_0^{9 \times n}$ | MNIST / SVHN |
| | mult | $\text{DIGIT}^2$ | $\text{INTEGER}_0^{81}$ | |
| | add-sub | $\text{DIGIT}^3$ | $\text{INTEGER}_{-9}^{18}$ | |
| | add-mod-3 | $\text{DIGIT}^2$ | $\text{INTEGER}_0^2$ | |
| | equal | $\text{DIGIT}^2$ | $\text{BOOL}$ | |
| | how-many-3-or-4 | $\text{DIGIT}^8$ | $\text{INTEGER}_0^8$ | |
| | not-3-or-4 | $\text{DIGIT}$ | $\text{BOOL}$ | |
| | less-than | $\text{DIGIT}^2$ | $\text{BOOL}$ | |
| | mod-2 | $\text{DIGIT}$ | $\text{INTEGER}_0^9$ | |
| NS-Leetcode | add-two-nums | $(\text{LIST}_2(\text{DIGIT}))^2$ | $\text{INTEGER}_0^{198}$ | MNIST / SVHN |
| | median-of-two-sorted-arrays | $(\text{LIST}_3(\text{DIGIT}))^2$ | $\text{DIGIT}$ | |
| | palindrome-number | $\text{LIST}_3(\text{DIGIT})$ | $\text{BOOL}$ | |
| | jump-game | $\text{LIST}_5(\text{DIGIT})$ | $\text{BOOL}$ | |
| | longest-consecutive-sequence | $\text{LIST}_5(\text{DIGIT})$ | $\text{INTEGER}_0^5$ | |
| | best-time-to-trade-stock | $\text{LIST}_8(\text{DIGIT})$ | $\text{DIGIT}$ | |
| | decode-ways | $\text{LIST}_3(\text{DIGIT})$ | $\text{INTEGER}_0^3$ | |
| | subsets-ii | $\text{LIST}_3(\text{DIGIT})$ | $\text{LIST}_8(\text{LIST}_3(\text{DIGIT}))$ | |
| | largest-rectangle | $\text{LIST}_6(\text{DIGIT})$ | $\text{INTEGER}_0^{54}$ | |
| | minimum-path-sum | $\text{LIST}_2(\text{LIST}_3(\text{DIGIT}))$ | $\text{INTEGER}_0^{36}$ | |
| | spiral-matrix-ii | $\text{DIGIT}$ | $\text{LIST}_9(\text{LIST}_9(\text{DIGIT})))$ | |
| | maximum-subarray | $\text{LIST}_5(\text{DIGIT})$ | $\text{INTEGER}_0^{45}$ | |
| | permutations-ii | $\text{LIST}_3(\text{DIGIT})$ | $\text{LIST}_6(\text{LIST}_3(\text{DIGIT}))$ | |
| | reverse-integer | $\text{LIST}_4(\text{DIGIT})$ | $\text{LIST}_4(\text{DIGIT})$ | |
| | longest-common-prefix | $\text{LIST}_3(\text{LIST}_3(\text{DIGIT}))$ | $\text{INTEGER}_0^3$ | |
| | longest-palindromic-substring | $\text{LIST}_5(\text{ALPHA})$ | $\text{LIST}_5(\text{ALPHA})$ | EMNIST |
| | longest-substring-without-repeat | $\text{LIST}_7(\text{ALPHA})$ | $\text{INTEGER}_0^7$ | |
| | reverse-string | $\text{LIST}_3(\text{ALPHA})$ | $\text{LIST}_3(\text{ALPHA})$ | |
| Custom | hand-written formula | $\text{LIST}_7(\text{SYMBOL})$ | $\text{FLOAT}$ | HWF |
| | $\text{sorting}_n$, where $n \in \{4, 8\}$ | $\text{LIST}_n(\text{DIGIT})$ | $\text{PERMUTAION}_n$ | MNIST |
| | severity of coffee leaf disease | $\text{LIST}_{10}(\text{BOOL})$ | $\text{INTEGER}_1^5$ | COFFEE |

Table 3: Characteristics of tasks in our benchmark suite.

coding problems, 2) neurosymbolic tasks from prior works including sorting and hand-written formula evaluation, and 3) a real-world task of determining disease severity of coffee leaves. In general, our benchmark suite is collected with the following desiderata:

1. **Challenging:** There should be a wide range of difficulty among tasks to demonstrate where ISED performs well and where it falls short. Tasks can be challenging due to having a large input space, a small output space, etc.
2. **Symbolic specification is natural to the subpart of the task:** The tasks should all contain both a neural component, which interprets the unstructured input data, and a symbolic component, which performs some computation using structured input data.
3. **Diversity of neural architectures:** The tasks should contain a wide variety of unstructured data and neural architectures to interpret the data.
4. **Diversity of symbolic components:** The tasks should include a wide variety of programs, including those that cannot be encoded by any differentiable logic programming language.

### 4.1 TASKS

A listing of tasks is provided in Table 3. For each task, we specify the potential configurations, the input and output structural mappings of the symbolic component, and the underlying perceptual dataset used. Below, we give a more detailed explanation on the benchmark.

**MNIST-R and SVHN-R.** MNIST-R, first proposed by Manhaeve et al. (2018) and extended by Li et al. (2023), contains several tasks operating on inputs of images of handwritten digits from the MNIST dataset (Lecun et al., 1998). This synthetic test suite includes tasks performing arithmetic (sum-2, sum-3, sum-4), comparison (less-than), counting (count-3, count-3-or-4), and negation (not-

3-or-4) over the digits depicted in the images. We also extend these tasks to use images from the Street View House Number (SVHN) dataset (Netzer et al., 2011), forming the SVHN-R dataset.

**NS-Leetcode.** To further experiment how the symbolic complexity affect learning performances, we curate a set of neurosymbolic tasks with the reasoning components collected from Leetcode problem solutions. By replacing symbolic task inputs with their unstructured counterparts, we obtain a set of 21 neurosymbolic Leetcode problems. Specifically, we replace the number inputs with digit image(s) from the MNIST (Lecun et al., 1998) and SVHN (Netzer et al., 2011) datasets. For numbers $\geq 10$, we use a list of numbers to represent the input. For string inputs, we employ the EMNIST dataset (Cohen et al., 2017). All datasets for tasks in NS-Leetcode have a training set (5K samples) and a testing set (500 samples), and are synthetically generated according to the input types.

**HWF and Sorting.** We include in our benchmark two tasks from prior works, namely hand-written formula (HWF) (Li et al., 2020) and sorting (Petersen, 2022). Most details of HWF and Sorting datasets are provided in Sec. 2. The HWF dataset contains 10K formulas of length 1-7, with 1K length 1 formulas, 1K length 3 formulas, 2K length 5 formulas, and 6K length 7 formulas. For Sorting, we evaluate on lists of length 4 and 8.

**Severity of Coffee Leaf Disease.** In this task, we use the dataset from Brito Silva et al. (2020) containing images of coffee leaves and bounding boxes of disease-affected regions. First, we use the Segment Anything Model (SAM) (Kirillov et al., 2023) to compute the total affected area for each leaf by segmenting the affected regions. These areas are then used to compute severity scores used as labels. The goal is for SAM to segment the images (without using labeled bounding boxes) for a CNN to subsequently classify regions as affected. The black-box program aggregates the areas of the regions classified as disease and returns a severity score.

## 5 EVALUATION

To demonstrate the effectiveness and applicability of ISED, we evaluate the algorithm on our benchmark suite. Our evaluation aims to answer the following research questions:

**RQ1** How does ISED compare to state-of-the-art neurosymbolic baselines in terms of accuracy?
**RQ2** Does ISED provide more interpretability than traditional neural approaches?

### 5.1 EVALUATION SETUP AND BASELINES

All of our experiments were conducted on a machine with two 24-core Intel Xeon CPUs, eight NVIDIA A100 GPUs, and 1.48 TB RAM. For tasks with MNIST or EMNIST dataset as unstructured data, we employ LeNet (Lecun et al., 1998), a 2-layer CNN based model. For SVHN, we use a similar CNN model but with 3 convolutional layers. For all MNIST-R, SVHN-R, and NS-Leetcode tasks, our model is trained for 10 epochs using Adam optimizer with learning rate 0.001. Unless otherwise noted, the sample count of ISED is set to 100 for all tasks. We report the average and standard deviation of accuracy obtained from 3 randomized runs.

For the task of Coffee Leaf Disease Severity, image segments produced by the Segment Anything Model (SAM) are scaled down and passed to a simple CNN based network for binary classification of whether there is a disease in the patch. During training, SAM is kept fixed and only the CNN classifier is trained. SAM also outputs fine-grained bit-masks which we use to compute the pixel area of each patch. As shown in Fig. 2, we sum the areas of the patches that are predicted as disease.

We pick as baselines Scallop (Li et al., 2023) and end-to-end neural CNN for the MNIST-R and SVHN-R tasks. For NS-Leetcode, we omit methods with a logic programming interface as baseline since most tasks in NS-Leetcode are unnatural to be written in logic programming languages. Further, we pick as baselines NGS and Scallop for HWF, DiffSort for sorting MNIST digits, and end-to-end CNN for Coffee Leaf Disease Severity.

### 5.2 RQ1: PERFORMANCE AND ACCURACY

To answer **RQ1**, we evaluate ISED's accuracy against those of the baselines. We note that while some of the baseline solutions can be written traditional logic programming languages, others such as DiffSort (Petersen et al., 2021a) are highly-tailored point solutions. ISED matches, and in many

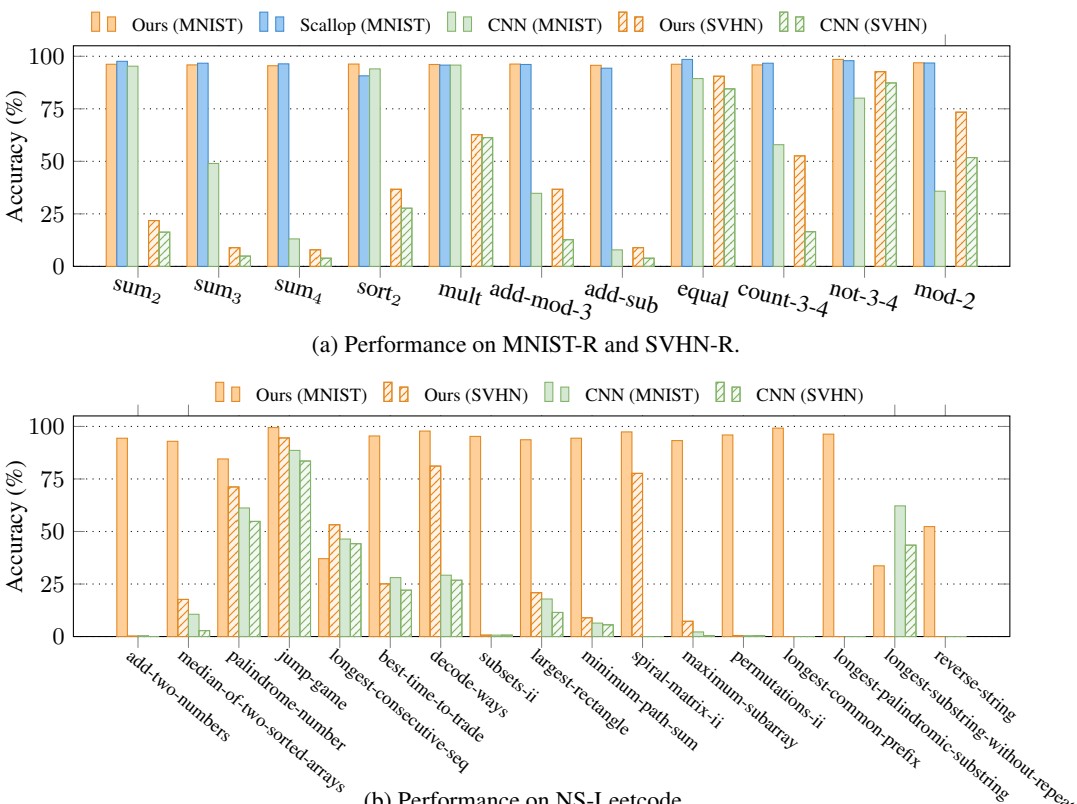

(a) Performance on MNIST-R and SVHN-R.

(b) Performance on NS-Leetcode.

Figure 3: Performance of ISED compared to selected baselines.

| Method | Accuracy (%) |
|---|---|
| NGS | **98.50** |
| NGS-MAPO | 71.70 |
| NGS-RL | 3.40 |
| Scallop | 97.85 |
| ISED (Ours) | 98.20 |

Table 4: Performance comparisons with NGS and Scallop on HWF.

| Method | Accuracy (%) | |
|---|---|---|
| | $n = 4$ | $n = 8$ |
| DiffSort (Odd-Even) | 94.9 | 87.9 |
| DiffSort (Bitonic) | 95.3 | 90.3 |
| ISED (Ours) | **95.6** | **92.6** |

Table 5: Performance on Sorting MNIST digits with $n \in \{4, 8\}$.

| Method | Accuracy (%) |
|---|---|
| CNN | 44.19 |
| ISED (Ours) | **60.47** |

Table 6: Performance comparisons on Severity of Coffee Leaf Disease.

cases surpasses, the accuracy of baselines simply by treating their programmatic solutions as black boxes. On average across all tasks, ISED achieved 30% higher accuracy than the neural baselines. Of the 11 benchmarks whose solution could be logically encoded, ISED beat Scallop on 6 of them with an average accuracy difference of 0.18%. Overall, as shown in Figure 3, ISED was the best performer on 16 out of 17 tasks in the benchmark, with an average accuracy difference of 30%.

We highlight two benchmarks to demonstrate the performance of ISED. For the HWF task, whose results are shown in table 4, ISED is within 0.3% accuracy of the highest performer NGS (Li et al., 2020); this attainment is despite the fact that, unlike NGS, ISED does not require encoding domain knowledge of the problem in a custom syntax and semantics. Furthermore, ISED can provide high accuracy in real-world scenarios against traditional approaches such as end-to-end neural; this point is exemplified in 6, where ISED beats a traditional CNN at coffee leaf disease image classification by over 16%. The high performance of ISED across these benchmarks, and many others, shows that ISED can achieve competitive accuracy despite higher generality and applicability.

## 5.3 RQ2: INTERPRETABILITY

To answer **RQ2**, we evaluate the interpretability of ISED on the Coffee Leaves Disease Severity task. In the pure neural approach, the model takes the entire leaf image as input and predicts a severity score as output, leaving decision-making to an uninterpretable black box. ISED's approach is more

Figure 4: Given a patch segmented from a coffee leaf image, our model is asked to predict whether the patch contains a leaf disease. Despite never received direct supervision on patch level labels, our method makes mostly correct predictions, providing interpretability and explanability.

granular, allowing for the integration of a black-box function that can assist a CNN in learning semantically meaningful representations. In the coffee leaf disease example, we define the black-box function to sum the area of only the segmentation masks that contain diseased spots and classify that sum according to thresholds. Therefore, we should expect our model to appropriately align with the specifications of the corresponding function's parameter and accurately classify whether or not a mask contains a diseased spot. In order to verify this, we manually inspected 44 patches from 10 different leaves and determined that our method produced 37 correct labels, indeed demonstrating the required semantic alignment. Sample leaves are shown in figure 4. This idea generalizes to any arbitrary black-box function; ISED provides a framework for networks in a neurosymbolic pipeline to conform to the interpretable specifications of the black-box function's parameters.

## 6 Related Work

**Neurosymbolic programming frameworks** These frameworks provide a general mechanism to define white-box neurosymbolic programs. DeepProbLog (Manhaeve et al., 2018) and Scallop (Li et al., 2023) abstract away gradient calculations behind a rule-based language. Others specialize in targeted applications, such as NeurASP (Yang et al., 2020) for answer set programming, or Neural-Log (Chen et al., 2021) for phrase alignment in NLP. ISED is similar to these approaches in that it seeks to make classes of neurosymbolic programs easier to write and access; however, it diverges by offering an interface not bound by any specific domain or language syntax.

**Specialized neurosymbolic methods.** The majority of the neurosymbolic learning literature pertains to point solutions for specific use cases Dutta et al. (2023); Wang et al. (2019). In the HWF example, NGS (Li et al., 2020), and several of its variants leverage a hand-defined syntax defining the inherent structure within mathematical expressions. Similarly, DiffSort (Petersen et al., 2021a) leverages the symbolic properties of sorting to produce differentiable sorting networks. Other point solutions address broader problem setups, such as NS-CL (Mao et al., 2019) which provides a framework for visual question answering by learning symbolic representations in text and images. For reading comprehension, the NeRd (Chen et al., 2021) framework converts NL questions into executable programs over symbolic information extracted from text. ISED aligns with all of these point solutions by aiming to solve problems that have thus far required technically specific solutions in order to access the advantages of neurosymbolic learning, but it takes an opposite and easier approach by forgoing significant specializations and instead leverages existing solutions as black boxes.

**Differentiable programming and non-differentiable optimization.** Longstanding libraries in deep learning have grown to great popularity for their ability to abstract away automatic differentiation behind easy-to-use interfaces. PyTorch (Paszke et al., 2019) is able to do so by keeping track of a dynamic computational graph. Similarly, JAX (Bradbury et al., 2018) leverages functional programming to abstract automatic differentiation. ISED follows the style of these frameworks by offering an interface to abstract away gradient calculations for algorithms used in deep learning, but ISED improves upon them by allowing systematic compatibility of non-differentiable functions.

## 7 Conclusion

We proposed ISED, a general framework for neurosymbolic learning with black-box programs. Unlike existing general neurosymbolic frameworks which require differentiable logic programs, ISED is compatible with Python programs and employs a sampling-based technique to approximate the gradient of the output of a program. We demonstrated that ISED achieves better accuracy than end-to-end neural models on synthetic and real-world benchmarks. ISED also achieves comparable or better accuracy compared to state-of-the-art general neurosymbolic frameworks.

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
