# OpenReview forum: "Beyond Differentiability: Neurosymbolic Learning with Black-Box Programs"
_ICLR.cc/2024/Conference — Submitted to ICLR 2024_

### Official Review · Reviewer_7YF3 · 2023-10-29

**Soundness:** 3 good
**Presentation:** 3 good
**Contribution:** 3 good
**Rating:** 6
**Confidence:** 2

**Summary:**

The authors of this paper introduce a new approach to neurosymbolic learning called Infer-Sample-Estimate-Descend (ISED). Neurosymbolic learning aims to combine classical algorithms and deep learning. Unlike existing neurosymbolic frameworks, ISED allows for the use of black-box programs written in general-purpose languages, expanding its applicability. ISED is designed for algorithmic supervision, where a black-box program is applied to the output of a neural model, and the goal is to optimize the model parameters using end-to-end labels. ISED consists of four phases: Infer, Sample, Estimate, and Descend, where neural models predict distributions for inputs, samples are generated, the program is executed, probabilities are estimated, and the loss function is computed. ISED is evaluated on 30 benchmark tasks with black-box programs written in Python and achieves higher accuracy than end-to-end neural approaches, often outperforming a state-of-the-art neurosymbolic framework called Scallop.

**Strengths:**

Good performance and quality experiences with clear text .

**Weaknesses:**

Poor quantitative aspects of training, including memory requirements, training time for each model, and for each dataset.

**Questions:**

What are the shortcomings related to the quantitative aspects of training, such as memory requirements, training time for each model, and for each dataset?

**Details Of Ethics Concerns:**

-

---

> ### Author Response · Authors · 2023-11-21
>
> There was a tradeoff between scalability and expressiveness in this work. ISED favors expressiveness, allowing the use of general-purpose languages such as Python, over scalability. Hence, we did not include memory requirements and training time for our benchmarks in this paper. For the sake of completeness, we will include these numbers in the final version, but we expect that the training time of ISED will generally be higher than the training time of Scallop.

---

### Official Review · Reviewer_9rTK · 2023-11-01

**Soundness:** 3 good
**Presentation:** 3 good
**Contribution:** 2 fair
**Rating:** 3
**Confidence:** 5

**Summary:**

This article introduces a Neuro-Symbolic learning framework designed to utilize structured knowledge pertaining to the outputs of neural networks, expressed through black-box programs. The method proposed in this paper requires only that the reasoning part be capable of forward reasoning, without necessitating that the output of the reasoning part be differentiable with respect to the input. Authors validate the adaptability of the learning framework through extensive experimentation on a variety of synthetic and classic benchmarks.

**Strengths:**

1.	The paper is easy to read, with Figures 1 and 2 providing a clear and intuitive illustration of the proposed method.
2.	The paper conducts a comprehensive evaluation of the proposed method on synthetic and classic tasks such as MNIST Add, HWF and Sorting, demonstrating the method’s adaptability.

**Weaknesses:**

1.	In the abstract, the authors mention that ‘existing general neuro-symbolic frameworks require that programs be written in differentiable logic programming languages’. However, there already exist frameworks aiming at bridging machine learning and logic reasoning such as Semantic Loss [1], Abductive Learning [2] and NEUROLOG [3], which do not impose a requirement for differentiability and they use non-differentiable programs. The paper does not conduct comparisons with such methods.
2.	The novelty of the proposed work needs to improve to meet the desired standards. The method proposed in this paper involves employing the REINFORCE algorithm to eliminate the requirement for differentiability in neural-symbolic system. However, this idea has already been introduced in the previous work [4]. Another critical component of the method, 'Estimate', fundamentally constitutes a sampling estimation of the well-known semantic loss [1], yet the paper does not provide reference to this work.

[1] Jingyi Xu, Zilu Zhang, Tal Friedman, Yitao Liang, and Guy Van den Broeck. A Semantic Loss Function for Deep Learning with Symbolic Knowledge, ICML 2018.

[2] Zhi-Hua Zhou. Abductive learning: Towards bridging machine learning and logical reasoning. Science China Information Sciences, 2019.

[3] Tsamoura, Efthymia, Timothy Hospedales, and Loizos Michael. Neural-Symbolic Integration: A Compositional Perspective, AAAI 2021.

[4] Cornelio Cristina, Jan Stuehmer, Shell Xu Hu, and Timothy Hospedales. Learning where and when to reason in neuro-symbolic inference, ICLR 2023.

3.	Inconsistent styles: some NeurIPS references include page numbers, while others do not; some conference names have abbreviations, while others do not.

**Questions:**

1.	Prolog is Turing-complete, possessing the same expressive capabilities as Python, and is also suitable for general-purpose use.
2.	The neural network's initial performance is nearly equivalent to a random output, and methods based on sampling may encounter difficulties in capturing certain symbols. How you address this cold start issue?

---

> ### Author Response · Authors · 2023-11-21
>
> Thank you for pointing out these references. We will cite them in the final version of this paper. The ideas in [1] and [4] seem similar, but our main contribution is in integrating sampling-based estimation of loss in the neurosymbolic learning pipeline. We are trying to implement the exact algorithm in [4] as a baseline for experimental evaluation, and will include it in the final version.

---

> > ### Comment · Reviewer_9rTK · 2023-11-23
> >
> > I have read the author responses.

---

### Official Review · Reviewer_Bw9k · 2023-11-01

**Soundness:** 2 fair
**Presentation:** 2 fair
**Contribution:** 2 fair
**Rating:** 5
**Confidence:** 2

**Summary:**

The paper presents a general Neuro-Symbolic solver approach using fixed black-box programs with the premise this allows anyone to write the program in any language as it removes the need for the program to be differentiable. This changes the neurosymbolic problem from program inference, to focus on parameter sampling and gradient propagation around the black box. The authors show that on three tasks  calculation, sorting and disease detection their method is able to outperform the baseline.

**Strengths:**

- In principle, the approach is a general-purpose neuro-symbolic approach.
- Despite the removal of gradients during the execution step, the performance is equivalent to the baseline
- The idea of using user-defined programs in execution is interesting and novel.
- The author's multi-dataset evaluation provides a broad context in different settings.  However, the leaf disease setting does not need to be a neuro-symbolic approach as shown by the simple program. A spatial reasoning test would probably have been a better choice.

**Weaknesses:**

- The introduction of the black-box programs seems to constrain the problem largely. In the writing, it isn't clear if the programs are only used as supervision or explicitly used as the symbolic aspect. If it is the latter this greatly reduces the difficulty of the problem as the symbolic aspect is largely unneeded, as you could include an expert program to solve without needing symbolic, this is especially evident in the leaf disease test as there are a number of off the shelf expert models that could be applied without needing a threshold of % diseased. The approach would have been better argued by another approach, such as logical reasoning, ideally on Knowledge Graphs, which would be a compatible setting to prior methods, increasing the number of comparisons the authors could perform.
- It isn't clear how this would scale to larger, more complex problems where the black-box program actually is complex. All examples are relatively trivial.
- It isn't clear how they handle the gradients around the black box as they just state an optimizer solves this without any specificity.

**Questions:**

- Greater explanation of whether the programs are learnt or if they only the inputs are being estimated.
- Explanation of how gradients are propagated
- Any prediction of how this would scale to complex problems

---

> ### Author Response · Authors · 2023-11-21
>
> ISED uses the programs for both supervision and the symbolic computation. In the leaf example, we calculate ground truth severity scores using the same program that we use as the symbolic aspect during training. To calculate the ground truth severity scores, we use human-labeled bounding boxes and corresponding areas calculated by SAM and pass them to the black-box program. Since all areas are labeled 1, the program simply returns the sum of the diseased areas in the bounding boxes. During training, we try to learn which of the segments returned by SAM contain diseased areas by using the same program as the symbolic component.
>
> We agree that most examples are relatively simple; we do not have benchmarks as complex as say, Sudoku, a popular task in the neurosymbolic literature. We also acknowledge that similar to prior work in neurosymbolic learning, there is a tradeoff between scalability and expressiveness in our work. ISED favors expressiveness, allowing the use of general-purpose languages such as Python, over scalability. That being said, we did include some tasks in the Leetcode benchmark suite that have large input spaces, as shown in Table 3 of the paper.
>
> We think it would be beneficial to add a benchmark where we can use the pure neural model to generate knowledge graphs, and the black box can just verify this reasoning. We will add a knowledge graph benchmark in the final version.
>
>
> **Q1:** The programs are not being learned, only the inputs are being estimated
>
> **Q2:** We do not compute gradients explicitly, but rather use a custom loss function that is made to reward certain inputs and penalize others.
>
> **Q3:** We still need to do more work to determine how ISED would scale to complex problems. We have found that simple tricks like increasing the sample count or moving from the add/mult to min/max computation in the loss can help solve problems that are more complex, such as HWF. However, we would need to find more complex benchmarks for testing ISED to answer this question.

---

### Official Review · Reviewer_4kqW · 2023-11-03

**Soundness:** 2 fair
**Presentation:** 3 good
**Contribution:** 2 fair
**Rating:** 5
**Confidence:** 2

**Summary:**

This work presents a general framework for neurosymbolic learning with black-box programs (ISED). This framework does not need the differentiability of the program and uses a sampling-based method to approximate the gradient of the program execution. The evaluation results show that ISED is more accurate.

**Strengths:**

1. This paper uses lots of illustrations, which make the presentation clear.

**Weaknesses:**

1. Limited Benchmark: The evaluation compares with Scallop and CNN. However, this work did not compare with another differentiating neurosymbolic program work: DeepProbLog, which limits the significance of the performance.
2. Scalability. This work can only cover an input length of 7, concluding the multiplication and additional operation. However, a traditional neurosymbolic program’s statement may not limit two these two operations.

**Questions:**

1. How accurate the sampling-based method is? Is there any theoretical analysis about the gradient error from the estimation?

---

> ### Author Response · Authors · 2023-11-21
>
> Prior work shows that Scallop achieves comparable or higher accuracy than DeepProbLog (DPL) on many simple arithmetic, comparison, and counting tasks involving MNSIT images ([1], Section 6.3, Figure 15 and Table 4). Because Scallop outperformed DPL on all these tasks, we did not include DPL as a baseline. We will make sure to add DPL experiments later for the sake of completeness.
>
> We acknowledge that similar to prior work in neurosymbolic learning, there is a tradeoff between scalability and expressiveness in our work. ISED favors expressiveness, allowing the use of general-purpose languages such as Python, over scalability. To this end, we used a modified version of the HWF task in our benchmarks which included only multiplication and addition operations to show that hand-written formulas, with a smaller input space, can still be learned under ISED.
>
> We don’t have any theoretical analysis for the sampling-based method yet.
>
> [1] Scallop: A Language for Neurosymbolic Programming. Ziyang Li, Jiani Huang, Mayur Naik. PLDI 2023

---

### Meta-Review · Area_Chair_PbPj · 2023-12-03

**Metareview:**

The paper proposes a framework for neurosymbolic learning with black-box programs that does not require differentiable logic programs, but uses a sampling-based technique to approximate the gradient of the output of non-differentiable programs/tools. While this is an interesting direction, the reviews present salient arguments about the suitability of this paper for ICLR in its current form. The most important ones are that of low originality and missing work. I fully agree. The concerns raised in the reviews need to be clarified before publication. We hope that the reviews are useful to you and hope you push for some of the next AI venues.

**Justification For Why Not Higher Score:**

Weak novelty, missing related work.

**Justification For Why Not Lower Score:**

N/A

---

### Decision · Program_Chairs · 2024-01-16

Reject